

# Non-invasive measurements to identify mungbean genotypes for waterlogging tolerance

PS Basavaraj[1], Krishna Kumar Jangid[1], Rohit Babar[1], Jagadish Rane[1,2], KM Boraiah[1], CB Harisha[1], Hanamanth Halli[1], Aliza Pradhan[1], Kuldeep Tripathi[3], K Sammi Reddy[1] and M Prabhakar[4]

[1] ICAR-National Institute of Abiotic Stress Management, Baramati, Baramati, India
[2] ICAR-Central Institute for Arid Horticulture, Bikaner, India
[3] ICAR-National Bureau of Plant Genetic Resources, New Delhi, India
[4] ICAR-Central Research Institute for Dryland Agriculture, Hyderabad, India

Corresponding author
PS Basavaraj, bassuptl@gmail.com

## ABSTRACT

As the best-fit leguminous crop for intercropping across time and space, mungbean promises to sustain soil health, carbon sequestration, and nutritional security across the globe. However, it is susceptible to waterlogging, a significant constraint that persists during heavy rains. Since the predicted climate change scenario features fewer but more intense rainy days. Hence, waterlogging tolerance in mungbean has been one of the major breeding objectives. The present experiment aimed to employ non-destructive tools to phenotype stress tolerance traits in mungbean genotypes exposed to waterlogging and estimate the association among the traits. A total of 12 mungbean genotypes were used in the present study to assess waterlogging tolerance at the seedling stage. Plant responses to stress were determined non-destructively using normalized difference vegetation index (NDVI) and chlorophyll fluorescence parameters at different time intervals. NDVI and grain yield were positively associated with control (r = 0.64) and stress (r = 0.59). Similarly, chlorophyll fluorescence (quantum yield of PS-II) also had a significant positive association with grain yield under both control (r = 0.52) and stress (r = 0.66) conditions. Hence, it is suggested that NDVI and chlorophyll fluorescence promise to serve as traits for non-destructive phenotyping waterlogging tolerance in mungbean genotypes. With the methods proposed in our study, it is possible to phenotype hundreds of plants for waterlogging tolerance efficiently.

## INTRODUCTION

Waterlogging in the agricultural field is increasingly emerging as a major production constraint in many crops, including mungbean (*Vigna radiata*) (*Amin et al., 2017*). Climate change projections reveal that episodes of waterlogging and flooding are likely to occur more often than ever (*Caretta et al., 2022*). Due to its short duration, it fits well into multiple cropping systems (*Kyu et al., 2021*). These systems successfully cover the field with a green canopy across space and time to trap atmospheric carbon and nitrogen. The magnitude of this crop's contribution can be judgedby the fact that it is cultivated in an

area of about 7.3 M ha with a production of 5.3 Mt around the world (*FAOSTAT, 2021*). India produces about 30% of the world's mungbean; however, during the pre-monsoon and monsoon seasons, waterlogging is a constant issue (*Nair & Schreinemachers, 2020*).

This crop is primarily produced in upland and lowland ecosystems as an intercrop with other legumes like pigeonpea, cereals like sorghum and maize, and oilseed crops like groundnut and sesame. (*Herridge et al., 2019*). Mungbean is mainly grown as a succeeding crop to rice either by broadcasting in the standing crop one week before rice harvest or by manual dibbling after harvest in the lowland ecosystems (*Gupta et al., 2016*). Excessive soil moisture before or after rice harvest exposes the seeds of the succeeding crop to waterlogging stress, which reduces germination and poor crop stand (*Zaman et al., 2018*).

When soil pores near the root zone are too saturated with water, waterlogging stress results. Due to oxygen deprivation, which reduces root respiration and adenosine triphosphate (ATP) generation and impairs root growth and function, it causes poor growth, development, and economic output (*Yamauchi et al., 2018*; *Kyu et al., 2021*). Further, a lack of oxygen causes abnormalities in the uptake and transfer of nutrients by roots. Additionally, anoxic conditions facilitate the development of microbes detrimental to crop plants (*Yu et al., 2022*). Some ions ($Mn2+$ and $Fe2+$) build up to potentially hazardous amounts due to prolonged waterlogging (*McKee & Mckevlin, 1993*).

Mungbean is susceptible to excess water, especially in the early growth stages. (*Bansal et al., 2019*; *Douglas et al., 2020*). Anoxic conditions due to waterlogging lead to poor nodulation and nitrogen fixation (*Singh & Singh, 2011*). Studies have also shown reduced growth of mungbean genotypes when exposed to waterlogging due to reduced gas exchange (*Ikram, Bhattarai & Walsh, 2022*).

To address the yield losses due to waterlogging stress, deeper insight into mechanisms of waterlogging tolerance is essential. Traits associated with waterlogging tolerance can help develop and deploy tolerant mungbean varieties. Waterlogging tolerance is complex, and plants have developed several tolerance mechanisms. These include altering morpho-physiological parameters by producing adventitious roots, altering the morphology of shoots and roots, and maintaining more significant levels of gas exchange. (*Barickman, Simpson & Sams, 2019*) and chlorophyll, a fluorescence parameter (*Smethurst & Shabala, 2003*). Since genetic variation in traits associated with waterlogging can be a base for further improvement of crops, it is essential to assess diverse mungbean genotypes on a large scale. Conventional phenotyping approaches such as evaluation and scoring for the waterlogging tolerance through visual evaluation or scoring (green *vs.* chlorosis portion of the shoot) often result in cumbersome and subjective experimental errors (*Walter, Studer & Kölliker, 2012*). Furthermore, destructive measures damage plants and make it difficult to do additional measurements. Hence, non-invasive phenotyping technologies are gaining immense importance in assessing genetic variation in stress responses of crop plants.

The normalized difference vegetation index (NDVI) is one trait that can help predict plant biomass, senescence, and plant health status by measuring reflectance variations in the spectrum's red and near-infrared areas. (*Verhulst & Govaerts, 2010*; *Jiménez et al., 2017*). High NDVI values are produced by a healthy plant's leaves, which absorb red light and reflect more near-infrared light. As a result, portable NDVI sensors were developed to
monitor plant health (*Verhulst & Govaerts, 2010*). These sensors are now frequently used to monitor the health status of the plants (*Verhulst & Govaerts, 2010*; *Jiménez et al., 2017*; *Lin et al., 2020*; *Kyu et al., 2021*). Likewise, the maximal quantum yield of photosystem II (PSII) can be measured by chlorophyll fluorescence kinetics non-invasively and has been widely employed for examining functional changes in the photosynthetic apparatus under various stresses in a range of photosynthetic tissues (*Lichtenthaler & Rinderle, 1998*; *Rane et al., 2019*; *Rane et al., 2021*). Hence, the objectives of the investigations were to assess the utility of non-destructive tools for differentiating waterlogging tolerance in mungbean genotypes based on their relationship with grain yield.

## MATERIALS & METHOD

### Experimental material

Twelve mungbean genotypes (Table 1) were selected based on the tolerance and sensitivity nature of these genotypes to waterlogging from the preliminary study (P. S. Basavaraj, 2021, unpublished data).

### Experimental location

An experiment was conducted in controlled conditions during the rainy seasons (June-September) of 2021 and 2022 at ICAR-National Institute of Abiotic Stress Management, Baramati, located in the Pune district of Maharashtra State, India (18°09′30.62″N, 74°30′03.08″E). Plants were grown in pots 12-inch-diameter filled with 13 kilograms of black clay loam soil. In brief, soil properties recorded were: pH 8.0, EC 0.24 dSm$^{-1}$, 164 kg N, 14 kg P, 139.65 kg K ha$^{-1}$, 65% clay, 27% sand, and 8% silt, soil bulk density was 1.33 g/cc and porosity were 54.2%. Before sowing, a recommended dose of fertilizers (12.5 kg/ha N + 25 kg/ha P2O5 + 12.5 kg/ha K2O) was applied. The amount was calculated on a soil weight basis and adequately blended into the soil. A total of seventy-two pots were prepared, and each genotype was sown in 10 pots (five for control and five for stress treatment) for five replications. Five seeds were sown in each pot; three plants were removed after establishment, and two healthy and uniform plants were kept in each pot.

### Stress imposition

Waterlogging stress imposition was carried out 20 days after seedling emergence when the first trifoliate leaf had fully opened. Stress was imposed by keeping five pots for each genotype in big pots with a diameter of 25 inches, which contained water, and the remaining pots (control plant) were irrigated regularly by maintaining 80% of field capacity. The water level in the stress treatment was maintained at 20 mm above the soil surface of experimental pots for eight days (*Ahmed, Nawata & Sakuratani, 2002*; *Kyu et al., 2021*). After the stress period, *i.e.,* on the ninth day, excess water was removed from the pots, and plants were allowed to recover.

### Traits measured

#### Total chlorophyll content

The amount of chlorophyll ($\mu$g/g of fresh weight) present in the mungbean leaves was estimated following the method of *Lichtenthaler & Wellburn (1983)*.

**Table 1  List and details of genetic resources of mungbean used in the present study.**

| Sl. NO | Genotype | Details |
|--------|----------|---------|
| 1. | EC-693356 | VC6153B–20P, Tolerant to deficit moisture, waterlogging tolerant |
| 2. | EC-693357 | Heat and salinity tolerant, waterlogging tolerant |
| 3. | EC-693358 | Heat tolerant, waterlogging sensitive |
| 4. | EC-693363 | moderately tolerant to waterlogging |
| 5. | Harsha | Heat and elevated CO2 tolerant, moderately tolerant to waterlogging |
| 6. | IC-415144 | Waterlogging tolerant |
| 7. | IPM-205-7 | Waterlogging sensitive |
| 8. | NM-94 | Resistant to mungbean yellow mosaic disease, sensitive to waterlogging |
| 9. | PAU-911 | Waterlogging tolerant |
| 10. | Vaibhav | Resistant to mungbean yellow mosaic disease, powdery mildew and stem fly, moderately tolerant to waterlogging |
| 11. | VC-3960-88 | Resistant to mungbean yellow mosaic disease, moderately tolerant to waterlogging |
| 12. | VC-6372(45-8-1) | Moderately tolerant to waterlogging |

## Quantum yield (Qmax)-(Fv/Fm)

Chlorophyll fluorescence in leaves was measured to study changes in maximum PSII efficiency in response to waterlogging. In brief, leaves were adapted to dark for 30 min before observations. The temperature was set at $25 \pm 1\,°C$ in the imaging chamber. The leaf images were captured at given time points by chlorophyll fluorescence measuring system (FC 1000-H/GFP, Handy Fluor Cam, P.S.I., Brno, Czech Republic) as described in *Nedbal et al. (2000)*. Fluorescence was detected by a high-sensitivity charge-coupled device (CCD) camera, and it was driven by FluorCam software package (FluorCam 7). Initially, the minimum fluorescence level (F0) of dark-adapted leaves was determined using non-actinic measuring flashes provided by super-bright light emitting diodes (LEDs) followed by a saturation pulse of light radiation ($2,500\,\mu mol\,(photon)\,m^{-2}\,s^{-1}$) to obtain the maximum fluorescence (Fm). The maximum photochemical efficiency of PSII (Fv/Fm) was calculated using the formula below (*Krause & Weis, 1991*).

$$Qmax = \frac{Maximal\ fluorescence\ (Fm) - Initial\ fluorescence\ (F0)}{Maximal\ fluorescence\ (Fm)}$$

For assessing response to waterlogging stress by image analysis, four colours were set manually for taking fluorescent images: blue (corresponding to Fv/Fm 0.8), yellow, green, and red (corresponding to Fv/Fm 0.1). Qmax was estimated at 25, 30, 35, 45, and 50 days after emergence (DAE).

## NDVI

The NDVI of mungbean plants was recorded at regular intervals at 25, 30, 35, 40, 45, 50, 55, and 60 DAE. NDVI was measured using a hand-held device (GreenSeeker®, Trimble, Westminster, CO, USA) from 1.0 m above the soil surface of the experimental pot by following *Verhulst and Govaerts's (2010)* method.

### Grain yield per plant (g)

Grain yield per plant was harvested at physiological maturity through manual harvesting, seeds were separated from pods by hand, and seeds were sun-dried to 13% moisture content. Finally, the weight of seeds from each plant was measured from each pot under stress and control treatment.

## Statistical analysis

The average data from each genotype were used to calculate the variance (ANOVA) analysis for the observed parameters to test mean differences between control and waterlogging treatment and among genotypes. Pearson's correlation coefficient determined the association between and among parameters. All the analysis was carried out using R software version R version 4.2.2.

## RESULTS

### Analysis of variance

The analysis of variance for the Qmax reading at different intervals and grain yield of mungbean genotypes under control and waterlogging environments were presented in Table 2, and ANOVA for NDVI at different intervals and grain yield in Table 3. The results of the ANOVA indicated that grain yield was significantly influenced by the treatment (normal and waterlogging), genotypes, and their interaction with the traits studied ($P < 0.05$). The experiment was conducted for two seasons, and analysis found that season interaction (years 2020 and 2021) was statistically insignificant ($P < 0.05$). The mean sum of squares due to Qmax reading at different intervals for genotypes, treatment, and their interaction was found significant ($P < 0.05$). Similarly, the mean sum of squares due to NDVI at different intervals was significant for the genotypes, treatments, and their interaction.

### Effect of waterlogging on NDVI, total chlorophyll content, the quantum yield of PS-II, and grain yield

In the present experiment, waterlogging for eight days at an early stage of the crop (20 days after sowing) reduced the growth and development (delayed flowering) and eventually yield (39.01%) of mungbean genotypes. Total chlorophyll content decreased (10%) among the genotypes under stress compared to the control. All the genotypes generally maintained significantly higher chlorophyll content under control than waterlogged plants, whereas EC-693356 retained significantly higher total chlorophyll content under stress. Genotypes such as EC-455144 and Vaibhav are on par with EC-693356 under stress conditions (Fig. 1).

NDVI (canopy greenness) varied significantly among control and waterlogging stress; due to waterlogging stress, canopy greenness was reduced (13.56%) under stress compared to control. Genotypes such as IC-415144, EC-693356, and PAU-911 are significantly superior to other genotypes under stress and control conditions for canopy greenness (Fig. 2).

The quantum yield of PS-II, measured through chlorophyll fluorescence, also varied across the treatment; it dropped by 14% under stress. Under normal conditions, all

**Table 2** Analysis of variance (ANOVA) for Qmax and grain yield of mungbean genotypes.

| Source of variation | DF | Mean sum of squares | | | | | |
|---|---|---|---|---|---|---|---|
| | | Qmax1 | Qmax2 | Qmax3 | Qmax4 | Qmax5 | GY |
| Factor A (genotypes) | 11 | 0.001** | 0.020** | 0.003** | 0.008** | 0.008** | 17.76** |
| Factor B (treatment) Control and waterlogging | 1 | 0.002** | 0.034** | 0.053** | 0.183** | 0.417** | 338.52** |
| Interaction (A × B) | 11 | 0.001** | 0.001** | 0.001** | 0.001** | 0.003** | 1.567** |
| Season 1 | 1 | 0.001** | 0.002** | 0.002** | 0.003** | 0.002** | 1.417** |
| Season 2 | 1 | 0.001** | 0.002** | 0.002** | 0.003** | 0.002** | 1.417** |
| S1 × S2 | 1 | NS | NS | NS | NS | NS | NS |

**Notes.**
Qmax1, Qmax2, Qmax3, Qmax4, and Qmax5 were taken at 25, 30, 35, 45, and 50 days after sowing (DAS), respectively, NS, Non-significant; GY, grain yield.
**Significance at 5% level of probability.

**Table 3** Analysis of variance (ANOVA) of mungbean genotypes for NDVI and grain yield of mungbean genotypes.

| Source of variation | DF | Mean sum of squares | | | | | | | | |
|---|---|---|---|---|---|---|---|---|---|---|
| | | NDVI1 | NDVI2 | NDVI3 | NDVI4 | NDVI5 | NDVI6 | NDVI7 | NDVI8 | GY |
| Factor A (Genotypes) | 11 | 0.004** | 0.005** | 0.004** | 0.004** | 0.006** | 0.006** | 0.012** | 0.019** | 17.76** |
| Factor B (Treatment) Control and waterlogging | 1 | 0.002** | 0.118** | 0.150** | 0.256** | 0.201** | 0.160** | 0.184** | 0.086** | 338.52** |
| Interaction (A × B) | 11 | 0.002** | 0.001** | 0.000** | 0.001** | 0.000** | 0.001** | 0.000** | 0.000** | 1.567** |
| Season 1 | 1 | 0.001** | 0.111** | 0.130** | 0.256** | 0.201** | 0.160** | 0.184** | 0.086** | 308.52** |
| Season 2 | 1 | 0.001** | 0.111** | 0.130** | 0.256** | 0.201** | 0.160** | 0.184** | 0.086** | 308.52** |
| S1 × S2 | 1 | NA | NA | NA | NA | NA | NA | NA | NA | NA |

**Notes.**
NDVI1, NDVI2, NDVI3, NDVI4, NDVI5, NDVI6, NDVI7, and NDVI8 were taken at 25, 30, 35, 40, 45, 50, 55, and 60 days after sowing, respectively; NS, non-significant; GY, grain yield.
**Significance at 5% level of probability.

the genotypes had similar Qmax values, while under stress conditions, IC-415144 and EC-693356 were significantly superior to other genotypes (Fig. 3).

There was a significant reduction in grain yield of mungbean genotypes in stress situations compared to optimum conditions. Yield reduction ranged from 27.97% in EC-693356 to 48.15% in EC-693358, with average yield reduction of 38.95% among all the genotypes. (Fig. 4).

## Relationship between NDVI, chlorophyll, Qmax, and grain yield

The relationship between grain yield, NDVI, and Qmax under optimum and waterlogging conditions was studied through correlation coefficient analysis, and results are presented in the form of a correlogram (Figs. 5 and 6). It is evident from the figure that NDVI readings at all the time intervals are significantly and positively associated with grain yield under both optimum and waterlogging conditions ($P < 0.05$). Similarly, Qmax had a significant positive relationship with grain yield under both conditions.

The further correlation coefficient between grain yield, NDVI, Qmax, and total chlorophyll content under control and waterlogging stress conditions was also estimated,

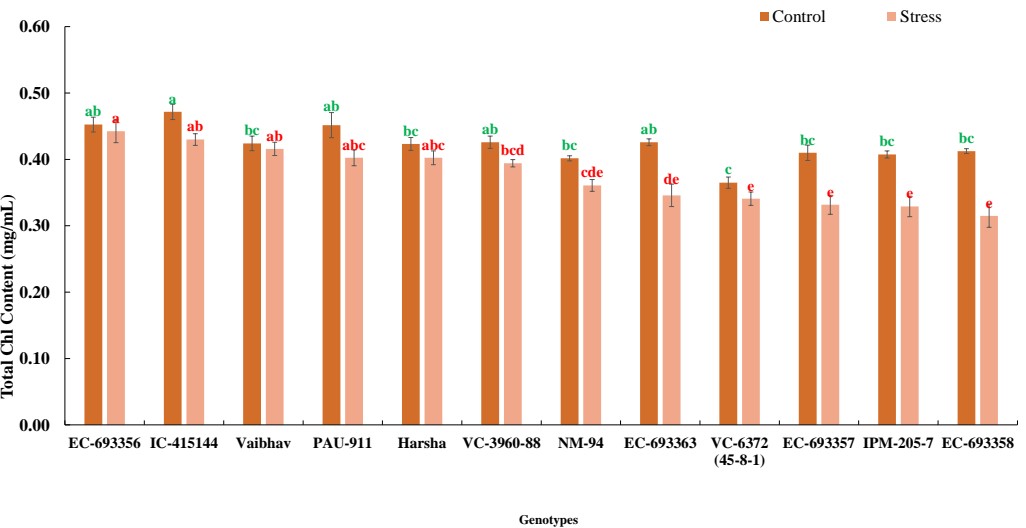

**Figure 1 Variation in total chlorophyll content of mungbean genotypes under control and waterlogging stress conditions.** The bars on the columns represent the SE, and different letters differ significantly by LSD ($p < 0.05$).

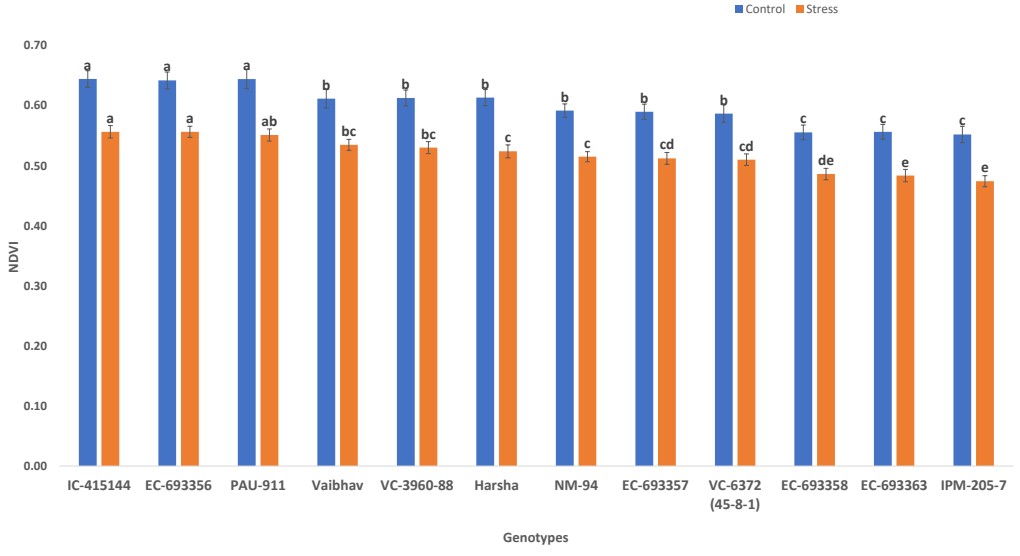

**Figure 2 Variation for canopy greenness (NDVI) of mungbean genotypes under control and waterlogging conditions.** The bars on the columns represent the SE, and different letters differ significantly by LSD ($p < 0.05$).

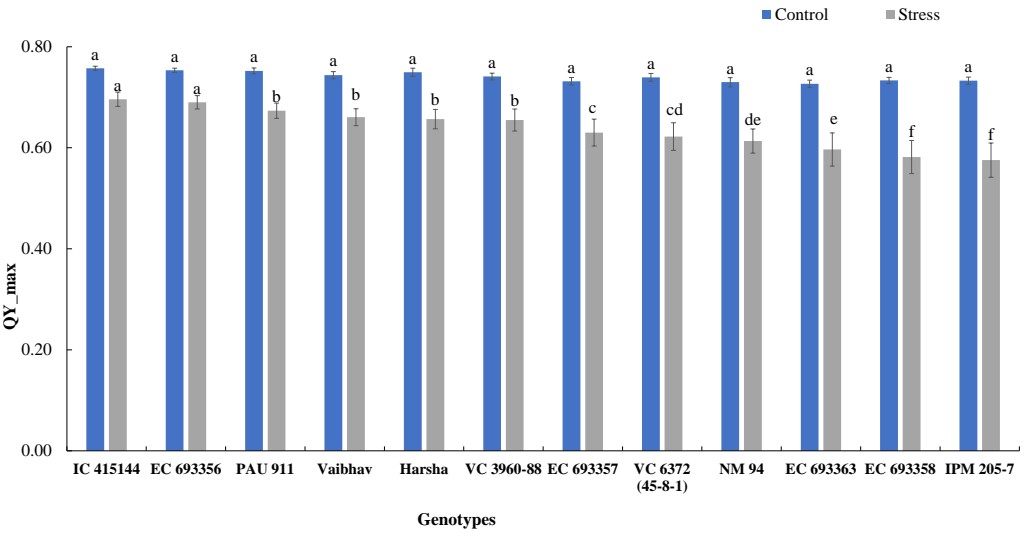

**Figure 3** **Variation in Qmax (quantum yield of PS-II) of mungbean genotypes under control and waterlogging stress.** The bars on the columns represent the SE, and different letters differ significantly by LSD ($p < 0.05$).

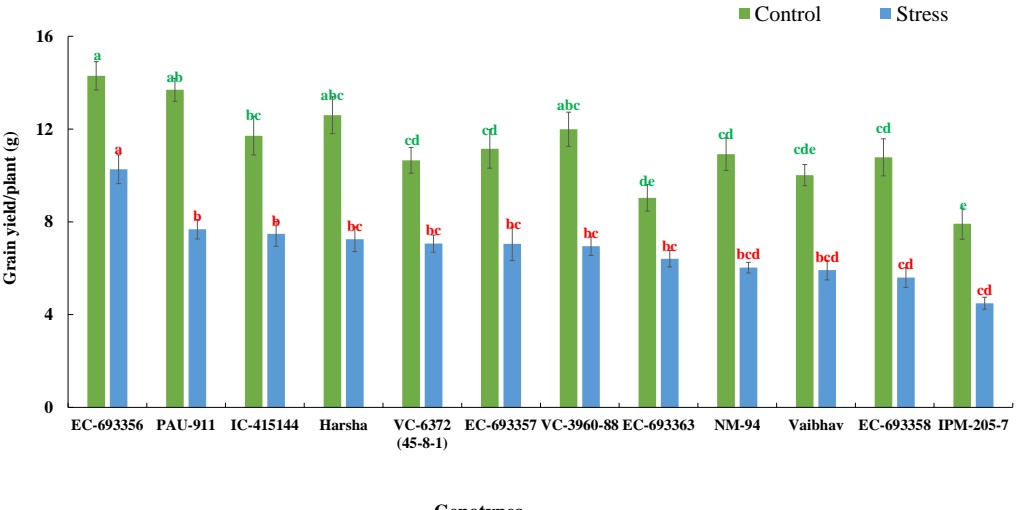

**Figure 4** **Variation in grain yield of mungbean genotypes under control and waterlogging stress conditions.** The bars on the columns represent the SE, and different letters differ significantly by LSD ($p < 0.05$).

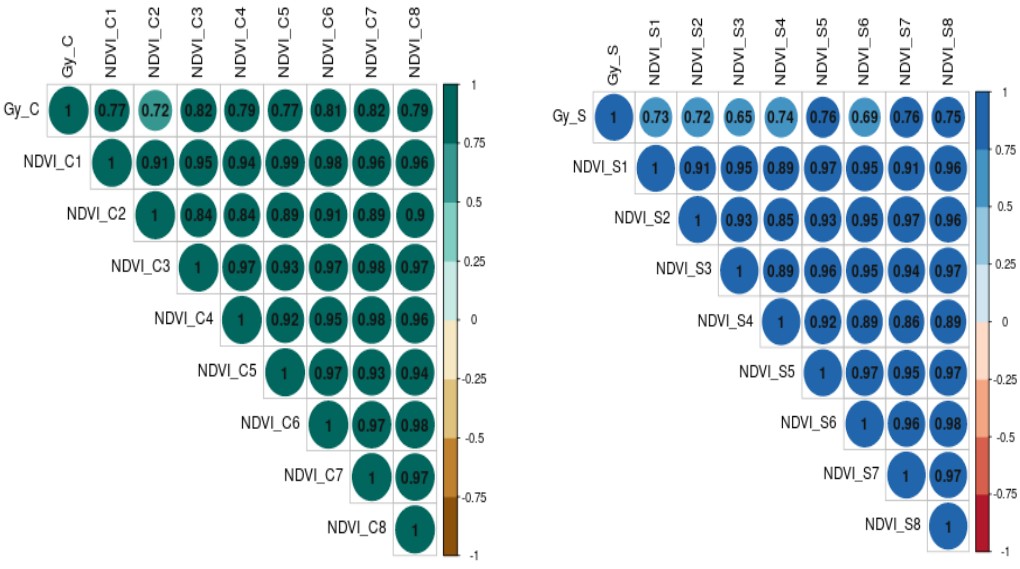

**Figure 5 The correlation coefficient between grain yield (Gy) and NDVI dynamics under control (C) and waterlogging stress (S) conditions, respectively.** The numbers (NDVI1, NDVI2, NDVI3, NDVI4, NDVI5, NDVI6, NDVI7, NDVI8) represent NDVI readings taken at 25, 30, 35, 40, 45, 50, 55, and 60 days after emergence, respectively.

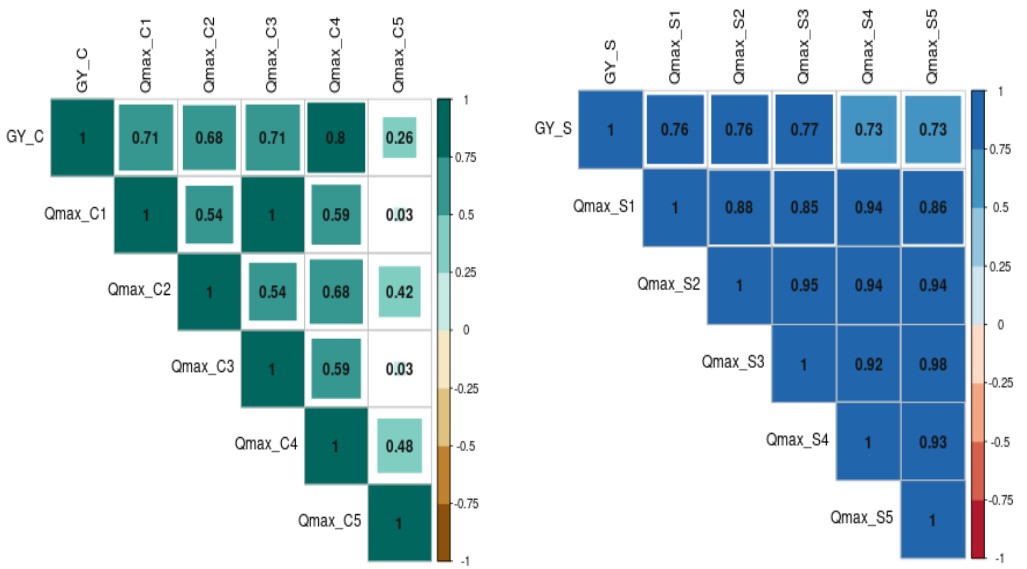

**Figure 6 The correlation coefficient between grain yield (Gy) and Quantum yield of PS-II (Qmax) dynamics under control (C) and waterlogging stress (S) conditions.** The numbers (Qmax1, Qmax2, Qmax3, Qmax4, Qmax5) represent Qmax readings taken at 25, 30, 35, 45, and 50 days after emergence, respectively).

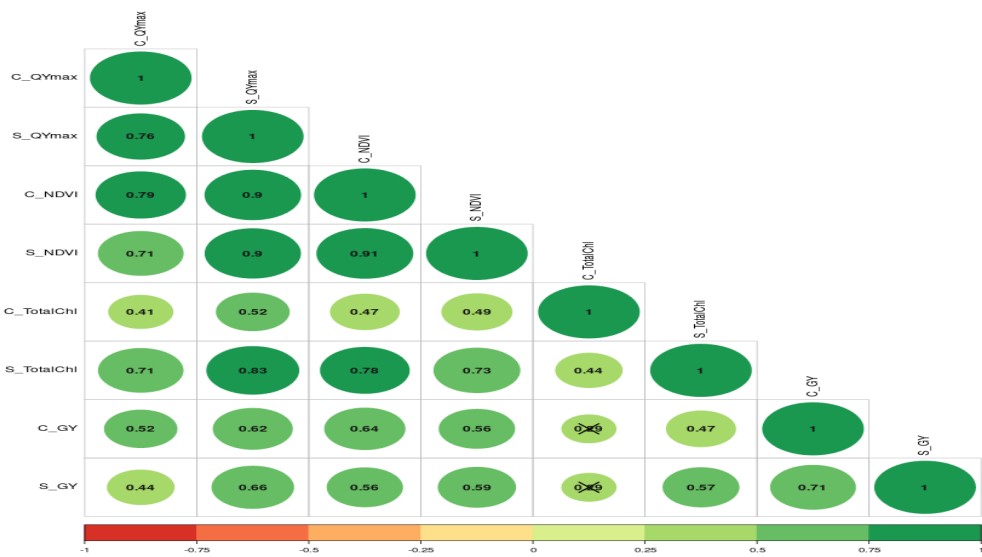

**Figure 7** The correlation coefficient between different parameters of mungbean genotypes under control and waterlogging stress conditions.

and results are presented in Fig. 7. Results revealed that C_NDVI ($r = 0.79$) and S_NDVI (0.71) also had a significant positive association with grain yield under control and stress, respectively. Similarly, Qmax_C and Qmax_S were significantly positively associated with grain yield under control and stress ($r = 0.52$ & 0.44). The relation between total chlorophyll content under normal ($r = 0.41$) and stress ($r = 0.71$) was also positive and significant.

# DISCUSSION

Waterlogging stress is a major production constraint for grain legumes such as mungbean (*Amin et al., 2017*), black gram (*Kyu et al., 2021*), pigeonpea (*Tyagi et al., 2023*), soybean (*Sathi et al., 2022*), which are highly vulnerable to waterlogging stress, particularly during early crop growth stages. Further, climate change models projected that crop productivity would decline more frequently due to waterlogging and other abiotic constraints (*Caretta et al., 2022*). One of the potential options to address this issue is the genetic improvement of mungbean for waterlogging tolerance. A more in-depth understanding of how different mungbean genotypes respond to waterlogging stress and tolerance mechanisms can offer clues regarding prospective traits to be selected in mungbean breeding programmes (*Davies, Turner & Dracup, 2000*). We employed a non-destructive physiological approach to study the association of grain yield with NDVI and PS-II efficiency of mungbean genotypes under control and waterlogging condition. A trait qualifies as the best selection criteria in a plant breeding program when it can differentiate the treatment, genotype, and treatment x genotype interaction in a way to identify promising genotypes as potential donors. In this

context, the utility of two traits that can be measured non-invasively has been discussed in the following section.

## Utility of NDVI in assessing the impact of waterlogging stress

The conventional way of evaluating and scoring for waterlogging tolerance is through visual evaluation or scoring (green *vs.* chlorosis portion of the shoot). However, these methods are subjective, often less accurate, and laborious (*Walter, Studer & Kölliker, 2012*). Because of these limitations, considerable interest emerged in deploying non-destructive techniques to assess how plants respond to abiotic stress. (*Barker et al., 2016*; *Lootens et al., 2016*). Additionally, deploying non-invasive phenotyping tools offers an opportunity to phenotype crop plants more precisely and robustly to achieve rapid genetic gain (*Walter, Studer & Kölliker, 2012*). The NDVI method estimates plant biomass and senescence by measuring reflectance variations in the spectrum's red and near-infrared areas (*Di Bella et al., 2004*; *Verhulst & Govaerts, 2010*). High NDVI values are produced by a healthy plant's leaves, which absorb red light and reflect more near-infrared light. Therefore, portable NDVI instruments were developed to expedite the assessment of plants' health state (*Verhulst & Govaerts, 2010*). These instruments are now often used to monitor crop health status under waterlogging stress. (*Jiménez et al., 2017*; *Lin et al., 2020*). In the present study, waterlogging on mungbean genotypes induced a series of changes in morpho-physiological traits such as yellowing, leaf necrosis, and defoliation of leaves at the morphological level. Further, changes in chlorophyll content, decreased canopy greenness, and reduced photosynthetic performance were noticed at the physiological level. NDVI, or greenness index, is an indicator that shows the greenness, density, and health of vegetation, which is directly associated with the chlorophyll content of plants. The higher the chlorophyll content, the higher the greenness and the higher the NDVI values (*Jiménez et al., 2017*; *Lin et al., 2020*). In the present experiment, we measured the canopy greenness at different intervals using NDVI, which was influenced by genotype, treatment, and their interaction. Results indicated that waterlogging stress has a differential effect on genotypes, and each genotype responds differentially to stresses.

The correlation coefficient between NDVI values and grain yield of mungbean was high at NDVI_S5 (45 DAS), NDVI_S7, and NDVI_S7 (55 & 60 DAS) under stress conditions demonstrating the method's potential for usage as a non-destructive phenotyping tool. This method could potentially be used to estimate the detrimental effects of waterlogging's on crop plants. However, the degree of association was less under stress, as evidenced by higher r values in optimum conditions. This is mainly due to decreased chlorophyll content under waterlogging conditions (*Kyu et al., 2021*). Further, the chlorophyll estimation results revealed a significant reduction (10%) in all the genotypes under stress conditions compared to the control (Fig. 1). Since there was less chlorophyll in the leaves in the early stages of crop growth *Akter et al. (2016)*, there were lower association values between seed yield and NDVI at the initial stages in both conditions.

Under stress conditions, there were low correlation values (NDVI_S1 to NDVI_S4) with a grain yield that is mainly due to the rapid degradation of chlorophyll content under waterlogging stress (*Kyu et al., 2021*; *Ikram, Bhattarai & Walsh, 2022*). As a result, the NDVI

has a broader range of applications for the non-destructive measurement of chlorophyll content and can reveal photosynthetic capability (*Lin et al., 2020*). Thus, it can be employed as a non-destructive screening approach for identifying waterlogging-tolerant mungbean genetic resources. Previously, *Jiménez et al. (2017)* and *Lin et al. (2020)* employed NDVI screen *Brachiaria* hybrids and summer squash for waterlogging stress tolerance.

## Utility of chlorophyll fluorescence in assessing PS-II response to waterlogging stress

In mungbean genotypes, waterlogging stress induced several physiological changes, including rapid chlorophyll degradation, decreased membrane stability, decreased stomatal conductance, and decreased photosynthetic rate (*Kyu et al., 2021*; *Ikram, Bhattarai & Walsh, 2022*). These modifications eventually impacted plant production, growth, and development. Additionally, waterlogging impairs photosynthesis's capacity to employ incoming photons, leading to photoinhibition, a concomitant decline in quantum yield, and a decrease in chlorophyll fluorescence (*Kumar et al., 2013*). Chlorophyll fluorescence is a non-destructive approach that has been extensively investigated for a range of photosynthetic tissues for investigating modifications in function in the photosynthesis machinery in response to several stresses. (*Lichtenthaler & Rinderle, 1998*; *Rane et al., 2019*; *Rane et al., 2021*). The first response of stressed plants is to close their stomata, which decreases the effectiveness of light and lowers ATP and NADPH consumption. As a result, this decrease in electron transfer causes a drop in PSII. A plant's photosystem II efficiency declines due to waterlogging, followed by a growth reduction (*Zheng et al., 2009*). We used a non-destructive physiological method to ascertain how waterlogging stress affected various mungbean genotypes' photosynthetic profiles and plant growth. We examined whether the plants could use these indicators as sensitive measures for estimating the photosynthetic capacity in the leaves corresponding to plant growth. The stress treatment influences the photosynthesis and yield of mungbean genotypes in the present study. Previous research has suggested that the PSII declines under waterlogging stress (*Smethurst & Shabala, 2003*; *Anee et al., 2019*). Like NDVI, we measured the association between the seed yield of mungbean and the quantum yield of PS-II at regular intervals, and the results found that Qmax had a positive and significant association with seed yield at all the interval readings under both control and stress conditions. Parallel to NDVI, higher r values between seed yield and Qmax were noticed under control conditions (Qmax_C4), *i.e., r* = 0.8. Along similar lines, there are also initially lower r values of Qmax with grain yield under control, and stress was observed when the root zone was saturated with excess water. Anoxia prevents aerobic respiration, which produces less energy. As roots are the main drivers in translocating water and nutrients, there will be imbalances in nutrients due to waterlogging and water translocation. This may also lead to a decrease in the solutes that enter the leaves through the transpiration stream. Since there is less $CO_2$ available for carbon fixation due to stomatal closure, internal $CO_2$ concentrations decrease. As a result, photosynthesis may also decline, which can induce senescence or even the death of plants. Lower r values of Qmax with grain yield under control and stress during later

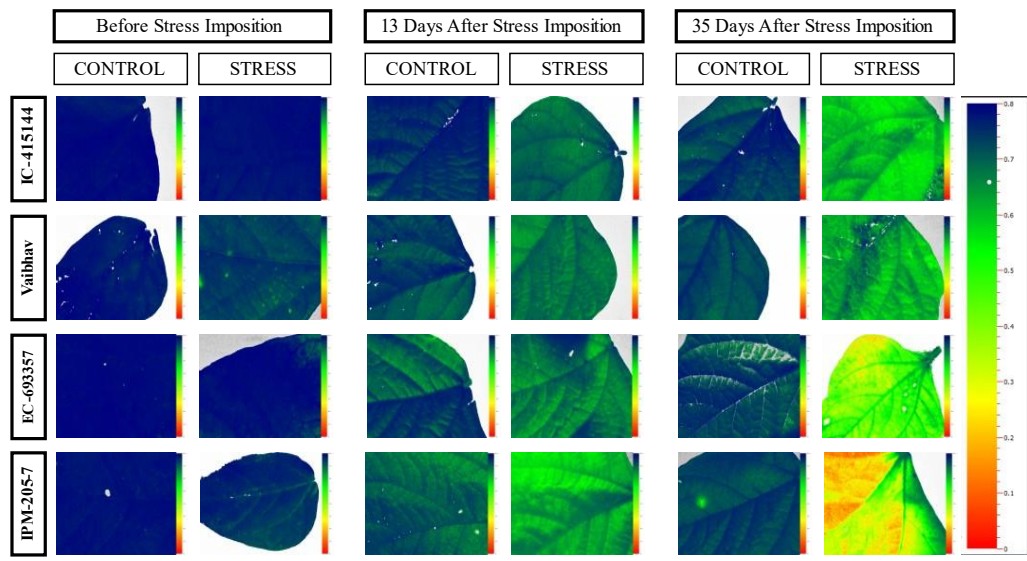

**Figure 8** Changes in quantum efficiency of PSII of the mungbean genotypes under control and waterlogging stress before and after stress.

stages are associated with leaf yellowing and maturity, which had already happened in the reproductive stage.

The quantum efficiency of PSII was reduced in all the mungbean genotypes. However, less reduction was noticed in EC-415144 and IC-693356 (Average of all the data points), indicating that these genotypes are superior to others in their ability to prevent photodamage (Fig. 8). According to *Zhu, Li & Shi (2016)*, the xanthophyll cycle's capacity to protect the photosynthetic apparatus from photoinhibition damage on exposure to waterlogging stress is the reason for this tolerance. Additionally, the decrease in PS-II was the first sign of waterlogging stress, making it useful as a preliminary screening criterion. A decline in Fv/Fm (Qmax) is an excellent indicator of photosynthetic degradation brought on by waterlogging stress, and the quantum efficiency of PSII (Fv/Fm) was once thought to be a proxy for the effects of environmental stress on photosynthesis. (*Murchie & Lawson, 2013*). Throughout our analysis, we noticed a reduction in Fv/Fm, which suggests that dark-adapted leaves may have PSII damage. These findings are consistent with a previous study that found a decrease in Fv/Fm in waterlogged mungbean (*Ahmed, Nawata & Sakuratani, 2002*; *Ikram, Bhattarai & Walsh, 2022*).

## Genetic variation of mungbean to waterlogging tolerance

We observed considerable genetic variations among mungbean genotypes for traits such as NDVI, quantum yield, total chlorophyll content, and grain yield under waterlogging and control conditions (Tables 2 and 3). Genotypes like EC-963356 could perform well under controlled and waterlogged conditions, while IC-415144, and Viabhav maintained significantly higher total chlorophyll content, followed by PAU-911 under waterlogging

**Table 4 Variation in chlorophyll content of mungbean genotypes under control and waterlogging stress.**

| Genotypes | Control | | | Stress | | |
|---|---|---|---|---|---|---|
| | ChlA | ChlB | TotalChl | ChlA | ChlB | TotalChl |
| EC-693356 | 0.40 | 0.06 | 0.45 | 0.33 | 0.12 | 0.44 |
| EC-693357 | 0.37 | 0.04 | 0.41 | 0.24 | 0.09 | 0.33 |
| EC-693358 | 0.26 | 0.15 | 0.41 | 0.25 | 0.06 | 0.32 |
| EC-693363 | 0.30 | 0.13 | 0.43 | 0.18 | 0.16 | 0.35 |
| Harsha | 0.37 | 0.06 | 0.42 | 0.34 | 0.07 | 0.40 |
| IC-415144 | 0.32 | 0.15 | 0.47 | 0.32 | 0.11 | 0.43 |
| IPM-205-7 | 0.27 | 0.13 | 0.41 | 0.23 | 0.10 | 0.33 |
| NM-94 | 0.36 | 0.04 | 0.40 | 0.30 | 0.06 | 0.36 |
| PAU-911 | 0.34 | 0.11 | 0.45 | 0.32 | 0.08 | 0.40 |
| Vaibhav | 0.28 | 0.14 | 0.42 | 0.26 | 0.16 | 0.42 |
| VC-3960-88 | 0.30 | 0.13 | 0.43 | 0.26 | 0.13 | 0.39 |
| VC-6372 (45-8-1) | 0.26 | 0.11 | 0.37 | 0.28 | 0.06 | 0.34 |
| Mean | 0.32 | 0.10 | 0.42 | 0.28 | 0.10 | 0.38 |
| Min. | 0.26 | 0.04 | 0.37 | 0.18 | 0.06 | 0.32 |
| Max. | 0.40 | 0.15 | 0.47 | 0.34 | 0.16 | 0.44 |
| Sd | 0.05 | 0.04 | 0.03 | 0.05 | 0.04 | 0.04 |

stress (Fig. 1). Canopy greenness (NDVI) was high in IC-415144, EC-963356, and PAU-911 relative to other genotypes under both controls, mainly due to the higher chlorophyll content (Fig. 2). Reduction in NDVI in other mungbean genotypes is mainly due to the rapid degradation of chlorophyll caused by the accumulation of ROS under stress (*Islam et al., 2019*). The higher yield of EC-963356 under stress could be attributed to chlorophyll retention and higher efficiency of PSII under stress, as revealed by maximum quantum efficiency (Table 4). A similar reduction in grain yield under waterlogging stress has been reported for mungbean genotypes due to a reduction in photosynthetic efficiency (*Kumar et al., 2013*).

## CONCLUSION

A two-year study clearly revealed a strong association of grain yield with NDVI and chlorophyll fluorescence-derived Qmax under both waterlogging stress and control conditions. These parameters could differentiate responses of genotypes to waterlogging treatments. It demonstrates the reliability of these non-invasive phenotyping tools and methods for phenotyping waterlogging tolerance in mungbean genotypes as they allow monitoring of plant growth and senescence patterns under waterlogged conditions over time. Hence, it is suggested that these traits can be efficiently employed for rapid and efficient phenotyping of a large number of genotypes for waterlogging tolerance. EC-963356 can be one of the potential genotypes that can be used as a donor for waterlogging tolerance in mungbean.

### Funding

This work was supported by the Indian Council of Agriculture Through Project "Phenotyping of Puleses for Enhaced Tolerance to Drought and Heat (NICRA)". The funders had no role in study design, data collection and analysis, decision to publish, or preparation of the manuscript.

### Grant Disclosures

The following grant information was disclosed by the authors:
The Indian Council of Agriculture Through Project "Phenotyping of Puleses for Enhaced Tolerance to Drought and Heat (NICRA)".

### Competing Interests

The authors declare there are no competing interests.

### Author Contributions

- PS Basavaraj conceived and designed the experiments, performed the experiments, analyzed the data, prepared figures and/or tables, authored or reviewed drafts of the article, and approved the final draft.
- Krishna Kumar Jangid conceived and designed the experiments, performed the experiments, analyzed the data, prepared figures and/or tables, and approved the final draft.
- Rohit Babar conceived and designed the experiments, performed the experiments, analyzed the data, prepared figures and/or tables, and approved the final draft.
- Jagadish Rane conceived and designed the experiments, analyzed the data, prepared figures and/or tables, authored or reviewed drafts of the article, and approved the final draft.
- KM Boraiah conceived and designed the experiments, performed the experiments, analyzed the data, prepared figures and/or tables, authored or reviewed drafts of the article, and approved the final draft.
- CB Harisha conceived and designed the experiments, performed the experiments, analyzed the data, authored or reviewed drafts of the article, and approved the final draft.
- Hanamanth Halli conceived and designed the experiments, performed the experiments, analyzed the data, authored or reviewed drafts of the article, and approved the final draft.
- Aliza Pradhan conceived and designed the experiments, authored or reviewed drafts of the article, and approved the final draft.
- Kuldeep Tripathi conceived and designed the experiments, authored or reviewed drafts of the article, and approved the final draft.
- K Sammi Reddy conceived and designed the experiments, authored or reviewed drafts of the article, and approved the final draft.

- M Prabhakar conceived and designed the experiments, analyzed the data, authored or reviewed drafts of the article, and approved the final draft.

## Data Availability

The raw data is available in the Supplementary File.

## Supplemental Information

Supplemental information for this article can be found online at http://dx.doi.org/10.7717/peerj.16872#supplemental-information.

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
