# Peer review of "Non-invasive measurements to identify mungbean genotypes for waterlogging tolerance"

_PeerJ, doi:10.7717/peerj.16872_

## Round 0.1 · original submission · Major Revisions

Dear Authors,
Please find the reviewers' comments below; address these changes and resubmit.

with regards

Reviewer 1 ·

Basic reporting

The authors evaluated NDVI and chlorophyll florescence as non-invasive parameters to evaluate waterlogged tolerance. The manuscript is well written and results are supported by statistical analysis.

Experimental design

Well designed.

Validity of the findings

No comment

Additional comments

However, some of the following points needs authors attention to further improve manuscript:
1. In the introduction section, line 37-39, the magnitude …… world, missing source citation.
2. Line 50-52 and 56-58 are need to revised and merged as both sentences have same meaning.
3. In line 71, state few examples of conventional phenotyping approach. However, they mentioned example in discussion section.
4. Not all genotypes response to waterlogged is given in table 1 while authors claim in line 90-91 that genotypes selected based on waterlogged sensitivity.
5. QY max and Qmax is same or different? In formula it is written as QY max while in other places Qmax is used.
6. In discussion, Qmax C and Qmax S are used and similarly for NDVI. Though it need to mentioned at first instance what does C and S stands (Control and Stress). Other places C and S used as prefix while other times they used as suffix.
7. It is not clear why they measured Qmax till 50 days while NDVI measured upto 60 days.

Reviewer 2 ·

Basic reporting

The study “Non-Invasive measurements to identify mung bean genotypes for water logging tolerance” carried out by Basavaraj et al. is of scientific importance and provides insight into the responses of 12 mung bean genotypes to water-logging stress.

The prime objective of the study is well defined and introduction section is presented in a well manner. However, I have few queries in introduction section.
1. Line number 38-39, provide source of Data/information?

Experimental design

1. What about check? Is there any check included or not in experimental material?
2. It would be better if the author also included other soil properties such as density and porosity of soil.
3. The study is lack in statistical analysis?
4. In present study why authors did not include any trait that would indicate tissue water status? What is the reason for this, can the author explain? In my opinion information about tissue water is very critical whenever a study conducted under water stress conditions.
5. When was the chlorophyll content in the plant recorded (how many days after seedling emergence), was the chlorophyll content recorded only once during crop growth or at regular intervals like NDVI and Qmax were recorded at regular intervals?
6. Why did the authors measure only total chlorophyll content? In my opinion, more specific results can be obtained if chlorophyll A and B content is measured instead of total chlorophyll.
7. Can the authors provide some images of the experiments specifically when the water was in logging conditions?

Validity of the findings

1. What is percent yield reduction in EC-963356 genotype during both the years of study?
2. In the present study only two major characters i.e., chlorophyll fluorescence and NDVI have been included which is a major limitation of this study. Furthermore, authors find EC-963356 genotype best during both the years of study. From perspective, considering any genotypes to be best based on only mean performance and with only two traits (CF and NDVI) is not justifiable. It would have been better if the authors included some more important physiological traits during the study.

Additional comments

No

Annotated reviews are not available for download in order to protect the identity of reviewers who chose to remain anonymous.

---

## Round 0.2 · Minor Revisions

Dear Authors,

Kindly check the following minor corrections:
1. Figures 1, 2, 3, and 4: labels on the X and Y axes were not clear and visible; increase the font size.
2. In figure 1: above the bars, ä", ''ab'', and 'abc' symbols represent what? For a better understanding of the readers, explain the significance in the caption of the figure.
3. Table and figures: the titles and captions should be modified with information about the number of individuals, mean, replication data that the figure represents, or the data used in that particular table.
4. Check the English language in the entire manuscript.

**Language Note:** The Academic Editor has identified that the English language must be improved. PeerJ can provide language editing services - please contact us at [email protected] for pricing (be sure to provide your manuscript number and title). Alternatively, you should make your own arrangements to improve the language quality and provide details in your response letter. – PeerJ Staff

Reviewer 1 ·

Basic reporting

NA as revised manuscript

Experimental design

NA as revised manuscript

Validity of the findings

NA as it is a revised manuscript

Additional comments

Authors s have addressed all the concerns very well.

Reviewer 2 ·

Basic reporting

The study “Non-Invasive measurements to identify mungbean genotypes for waterlogging tolerance” by Basavaraj et al. is of scientific importance and provides insight into the responses of 12 mungbean genotypes to water-logging stress.

The authors have included the suggestion and I am satisfied with the response from the authors

Experimental design

Well the authors answered most of the queries, however I urge to authors to include the following points
1. Mention the tolerant as well as susceptible check in the material used section
2. Please mention in MS when (at what stage/how many days after sowing/transplanting) and how many times each trait was recorded
3. What is the sample size for each trait?
4. I would still like the authors to mention chlorophyll A and B separately with percentage reduction so that it can be understood how the water logging stress affects the chlorophyll dynamics in mungbean
5. Mention the significance level (p < 0.05 or < 0.01) in MS for a better understanding of readers

Validity of the findings

Please include the suggestions

---

## Round 0.3 · accepted · Accept

Congratulations to the authors!
Significant improvements were made in the revised manuscript. Therefore, it is recommended for publication in PEERJ.